# Efficacy and safety of dofetilide and sotalol in patients with hypertrophic cardiomyopathy

Chris Chen [1], Mallika Lal[1], Yunwoo Burton[1], Hongya Chen[1], Eric Stecker[1], Ahmad Masri[1] & Babak Nazer[2✉]

## Abstract

**Background** Professional society practice guidelines conflict regarding their recommendations of dofetilide (DOF) and sotalol (STL) for treatment of arrhythmias in hypertrophic cardiomyopathy (HCM), and supporting data is sparse. We aim to assess safety and efficacy of DOF and STL on arrhythmias in HCM.

**Methods** This was an observational study of HCM patients treated with DOF or STL for atrial fibrillation (AF) and ventricular arrhythmias (VA). Outcomes of drug discontinuation and arrhythmia recurrence were compared at 1 year and latest follow-up by Kaplan–Meier analysis. Predictors of drug failure were studied using uni- and multi-variable analyses. Drug-related adverse events were quantitated.

**Results** Here we show that of our cohort of 72 patients (54 ± 14 years old, 75% male), 21 were prescribed DOF for AF, 52 STL for AF, and 18 STL for VA. At 1 year, discontinuation and recurrence rates were similar for DOF-AF (38% and 43%) and STL-AF (29% and 44%) groups. Efficacy data was similar at long-term follow-up of 1603 (IQR 994–4131) days, and for STL-VA. Drug inefficacy was the most common reason for discontinuation (28%) followed by side-effects (13%). Incidences of heart failure hospitalization (5%) and mortality (3%) were low. One STL-AF patient developed non-sustained torsades de pointes in the setting of severe pneumonia and acute kidney injury, but there were no other drug-related serious adverse events.

**Conclusions** DOF and STL demonstrate modest efficacy and satisfactory safety when used for AF and VA in HCM patients.

## Plain language summary

Hypertrophic cardiomyopathy (HCM) is a genetic condition that affects the heart muscle by making it abnormally thick. It often also causes abnormalities in the heartbeat, known as arrhythmias, which can cause symptoms such as dizziness and shortness of breath, or death. Historically it has been advised that some drugs that can affect the heartbeat should not be used in those with HCM, leaving people with HCM to be treated with other drugs that have undesirable side effects. We studied HCM patients who had been prescribed two of the drugs that were advised not to be used, called dofetilide and sotalol. The drugs were found to have been safe and effective over a 4-year period. These results suggest that clinical guidelines should be updated to support the use of these drugs for the treatment of arrhythmias in patients with HCM.

[1] Knight Cardiovascular Institute, Oregon Health & Science University, Portland, OR, USA. [2] UW Medicine Heart Institute, University of Washington, Seattle, WA, USA. ✉email: bnazer@cardiology.washington.edu

Hypertrophic cardiomyopathy (HCM) is an autosomal dominant disorder of the sarcomere characterized by myocyte hypertrophy, myocyte disarray, and interstitial fibrosis[1]. HCM is well-known to be associated with increased risk for atrial fibrillation (AF), ventricular arrhythmia (VA), heart failure (HF), and cardiac arrest[2–6]. HCM patients poorly tolerate AF, and thus a rhythm control strategy is often required[7]. The 2014 AF guidelines gave amiodarone and disopyramide a Class IIA recommendation for medical management, but discouraged the use of dofetilide (DOF) and sotalol (STL) for severe left ventricular hypertrophy with wall thickness >15 mm (this remained unchanged with the 2019 AF guidelines update)[8,9]. Amiodarone's long-term toxicities pose management dilemmas especially for younger HCM patients, and disopyramide is primarily used for its negative inotropic effects to improve symptoms with limited data on its anti-arrhythmic efficacy[10].

Observational studies comparing amiodarone with DOF and STL in this population have suggested no increased DOF- or STL-related mortality[11]. The 2020 HCM guidelines currently deem use of DOF and STL as "reasonable" first-line medical treatment for AF[12,13], but many clinicians remain skeptical. Existing sparse data regarding efficacy and safety of DOF and STL for AF in HCM demonstrates moderate efficacy and reassuringly low rates of drug-related adverse events[14,15]. There are no dedicated studies of STL for HCM VA, and STL usage is largely extrapolated from studies using amiodarone for predominantly non-sustained VT[16–18]. We aimed to study safety, efficacy, and clinical outcomes of DOF and STL in a larger cohort of HCM patients with AF and/or VA. In 72 patients who took DOF or STL for treatment of arrhythmias in HCM, modest efficacy and satisfactory safety were found over a 4-year period.

## Methods

**Data collection**. We conducted a retrospective cohort study of HCM patients treated with DOF or STL for AF and/or VA from 2001 to 2021 at Oregon Health and Science University. IRB (OHSU Multidisciplinary Ventricular Arrhythmia Program Registry) approval was obtained, waiving informed consent due to the retrospective and otherwise anonymous nature of this study. We searched our institution's electronic medical record for patients with a diagnosis of "hypertrophic cardiomyopathy" (obstructive or non-obstructive including all relevant ICD-9 and ICD-10 codes within problem list and past medical history), and then filtered them by searching for "dofetilide" or "sotalol" on current or prior medication lists. Four patient groups were collected according to arrhythmia and medication: DOF-AF, STL-AF, STL-VA, and STL-All (both AF and VA). Manual chart review confirmed correct diagnoses of HCM not just by written documentation but also corroborated by echocardiography or cardiac MRI. Patients were excluded if found with erroneous or mislabeled diagnoses, age less than 18 years at time of drug initiation, and for lack of documentation regarding drug loading, arrhythmia recurrences, or follow-up. Baseline characteristics, pertinent baseline parameters of echocardiography, baseline ECG parameters prior to drug initiation as well as complications during drug initiation were noted. Normal cutoffs for left ventricular ejection fraction (LVEF), interventricular septal wall thickness in diastole (IVSd), left posterior wall thickness in diastole (LPWd), left atrial (LA) size and left ventricular outflow tract (LVOT) gradients ≥30 mmHg were appropriated according to American Society of Echocardiography guidelines. Systolic anterior motion (SAM) of the anterior mitral valve leaflet was defined based on mentioning in the echo report from qualitative assessment by the echo reader and not re-adjudicated by us. Baseline ECG parameters included corrected QT (QTc) and QRS

duration (QRSd). Duration of follow-up was calculated from start of drug initiation to date of chart review. Duration of treatment was calculated from start of drug initiation to drug discontinuation or date of chart review if still actively taking. Primary outcomes included rates of drug discontinuation and incidences of arrhythmia recurrence at 1-year post drug initiation and at time of latest chart review follow-up. Reasons for discontinuation were compiled and organized into adverse events and side effects. Adverse events included HF hospitalization, QT prolongation, hypotension, and bradyarrhythmia. Side effects included fatigue, dyspnea, weakness, dizziness, GI intolerance, hives, and depression. Arrhythmia recurrence was determined by evidence of such on ECG, extended cardiac monitoring, cardiac implantable device interrogations, admissions for cardioversions or ablations with active arrhythmia, or any documentation in the electronic medical record declaring recurrence. VA was defined as sustained ventricular tachycardia, ventricular fibrillation, or requiring appropriate antitachycardia pacing or defibrillations. Instances where STL was used solely for non-sustained ventricular tachycardia or premature ventricular complexes were excluded. Clinical outcomes included HF hospitalizations, cardiovascular death, and all-cause mortality at 1 year from drug initiation.

**Statistics and reproducibility**. For descriptive statistics, continuous variables are presented as means with standard deviation or medians with inter-quartile range (IQR); categorical variables are expressed as frequencies (percentage). To compare baseline characteristics between DOF AF and STL AF group, student's t-test were used if normally distributed and Wilcoxon rank sum test if not for continuous variables. Pearson's chi-square test or Fisher's exact test were used for categorical variables as appropriate. Kaplan–Meier curves were generated for estimated rates of drug discontinuation and arrhythmia recurrence in DOF and STL groups at 1 year after drug initiation. Due to differences in follow-up periods between drugs, we artificially censored at 7 years. Univariable and multivariable cox proportional hazard models were computed to identify factors associated with drug discontinuation and arrhythmia recurrence. All statistical tests were two-sided with $P < 0.05$ considered statistically significant. All statistical analyses were conducted using R Statistical Software version 4.0.5 (R Core Team 2020, R Foundation for Statistical Computing, Vienna, Austria).

**Reporting summary**. Further information on research design is available in the Nature Portfolio Reporting Summary linked to this article.

## Results

**Baseline characteristics**. A total of 122 patients were isolated based on initial search criteria from the medical record, and 72 patients met inclusion criteria after manual chart review. There were 21 DOF and 52 STL HCM patients with AF, 18 STL patients with VA. Within these groups were 10 patients on both DOF and STL at separate times, and 9 patients on STL who had both AF and VA. Mean ages at drug initiation were $53 \pm 14$ years in the DOF-AF group and $55 \pm 16$ years in the STL-AF group. Sixteen (76%) patients were male in the DOF-AF group, and 31 (60%) patients were male in the STL-AF group ($p = 0.18$). Presence of hypertension, hyperlipidemia, diabetes, stroke, venous thromboembolism, obstructive sleep apnea, tobacco smoking, chronic kidney disease, obstructive coronary artery disease, types of medications, and creatinine values were similar between DOF AF and STL AF groups (Table 1). There were 5 (24%) DOF-AF patients, 11 (21%) STL-AF patients, and 6 (33%) STL-VA patients who had prior septal myectomy. At the time of drug

**Table 1 Baseline patient characteristics at the time of medication initiation.**

| | Dofetilide – AF (N = 21) | Sotalol – AF (N = 52) | Sotalol – VA (N = 18) | Sotalol – All (N = 61) | P-value (Dofetilide – AF vs Sotalol – AF) |
|---|---|---|---|---|---|
| Age (years) | 53.7 ± 14.2 | 54.5 ± 15.5 | 49.3 ± 13.0 | 53.6 ± 15.0 | 0.639 |
| Male, N (%) | 16 (76.2) | 31 (59.6) | 15 (83.3) | 40 (65.6) | 0.181 |
| White race, N (%) | 21 (100) | 45 (86.5) | 11 (61.1) | 49 (80.3) | 0.182 |
| Hypertension, N (%) | 12 (57.1) | 27 (51.9) | 11 (61.1) | 31 (50.8) | 0.696 |
| Hyperlipidemia, N (%) | 9 (42.9) | 31 (59.6) | 11 (61.1) | 35 (57.4) | 0.193 |
| Diabetes, N (%) | 2 (9.5) | 10 (19.2) | 4 (22.2) | 14 (23.0) | 0.489 |
| Obstructive Sleep Apnea, N (%) | 11 (52.4) | 26 (50.0) | 9 (50.0) | 31 (50.8) | 0.854 |
| Tobacco Smoking, N (%) | 2 (9.5) | 9 (17.3) | 1 (5.6) | 9 (14.8) | 0.494 |
| Chronic Kidney Disease, N (%) | 2 (9.5) | 4 (7.7) | 3 (16.7) | 5 (8.2) | 0.454 |
| Congestive Heart Failure, N (%) | 5 (23.8) | 17 (32.7) | 5 (27.8) | 17 (27.9) | 0.513 |
| NYHA Class I | 15 | 33 | 13 | 42 | |
| NYHA Class II | 4 | 12 | 4 | 12 | |
| NYHA Class III | 2 | 7 | 1 | 7 | |
| NYHA Class IV | 0 | 0 | 0 | 0 | |
| Obstructive CAD, N (%) | 4 (19.0) | 10 (19.2) | 3 (16.7) | 9 (14.8) | 1 |
| Septal Myectomy, N (%) | 5 (23.8) | 11 (21.2) | 6 (33.3) | 14 (23.0) | 0.905 |
| Alcohol Septal Ablation, N (%) | 0 (0.0) | 5 (9.6) | 2 (11.1) | 5 (8.2) | 0.508 |
| Presence of an ICD, N (%) | 9 (42.9) | 31 (59.6) | 18 (100) | 40 (65.6) | 0.193 |
| Presence of PPM, N (%) | 0 (0) | 1 (1.9) | 0 (0) | 1 (1.6) | 1 |
| Ablation, N (%) | 12 (57.1) | 11 (21.2) | 4 (22.2) | 13 (21.3) | 0.003 |
| Electrical Cardioversion, N (%) | 16 (76.2) | 32 (61.5) | 4 (22.2) | 34 (55.7) | 0.232 |
| Paroxysmal AF, N (%) | 1 (4.8) | 7 (13.5) | 3 (16.7) | 7 (11.5) | 0.425 |
| Persistent AF, N (%) | 20 (95.2) | 42 (80.8) | 4 (22.2) | 42 (68.9) | 0.16 |
| Permanent AF, N (%) | 0 (0) | 0 (0) | 1 (5.6) | 1 (1.6) | |
| Atrial Flutter, N (%) | 4 (19.0) | 11 (21.2) | 2 (11.1) | 11 (18.0) | 1 |
| History of Sustained VT, N (%) | 1 (4.8) | 10 (19.2) | 15 (83.3) | 18 (29.5) | 0.16 |
| Creatinine (mg/dL) | 1.0 ± 0.2 | 1.0 ± 0.3 | 1.1 ± 0.4 | 1.0 ± 0.3 | 0.873 |

*NYHA* New York Heart Association, *CAD* coronary artery disease, *ICD* implantable cardioverter-defibrillator, *PPM* pacemaker, *AF* atrial fibrillation, *VT* ventricular tachycardia.

initiation, 9 (43%) patients in the DOF-AF and 31 (59.6%) patients in the STL-AF group had implantable cardioverter defibrillators (ICD) ($p = 0.193$). All 18 patients in the STL-VA group had ICDs prior to drug initiation. There were more patients in the DOF-AF than STL-AF group with prior ablations (57% vs 21%, $p = 0.003$). The majority of AF was classified by medical record documentation as persistent. Total daily DOF dose was 750 ± 264 mcg, and STL dose was 210 ± 81 mg. Drug doses are detailed in Supplementary Table 1.

**Imaging and ECG baseline characteristics**. Summary of baseline parameters of echocardiography and ECG are seen in Table 2. Of note IVSd was thinner in the DOF-AF group (1.4 ± 0.4 cm) than in STL-AF (1.7 ± 0.6 cm, $p = 0.007$). LA size by LA diameter in the parasternal long axis view and LA volume index by Simpson's biplane method were mostly at least moderately elevated. Within the STL-VA population, only 8 (44%) patients had CMR of which 5 of these did not quantify late gadolinium enhancement (LGE). In this same group, 3 (17%) had LV aneurysms noted, and 3 (17%) had LVEF < 50% at time of drug initiation. All 21 patients on DOF were loaded inpatient while 32 (62%) STL-AF and 11 (61%) STL-VA patients were loaded inpatient. QTc was similarly normal across all groups.

**Efficacy outcomes and predictors**. Total follow-up days for all patients were a median of 1603 (IQR 994–4131) days. Total days on treatment for DOF AF and STL AF groups were similar with

medians of 563 (IQR 96, 1362) and 566 (IQR 165, 1321), respectively ($p = 0.514$). At 1 year, drug was discontinued in 8 (38%) DOF-AF and 15 (29%, $p = 0.358$) STL-AF patients, respectively, with no difference in discontinuation rates between groups ($p = 0.223$; Fig. 1). Inefficacy (arrhythmia recurrence) was the most common reason for discontinuation: 6 (29%) in DOF-AF group, 14 (27%) in STL-AF group, 3 (17%) in the STL-VA group, followed by side-effects (Table 3).

At 1 year, 9 (43%) DOF-AF and 23 (44%) STL-AF patients had arrhythmia recurrence ($p = 0.665$), with a trend towards more arrhythmia recurrence in the DOF group over time ($p = 0.084$; Fig. 1). STL-VA patients had 50% arrhythmia recurrence at 1 year.

Multivariable analysis (Table 4) demonstrated that increasing age, increased IVSd thickness, presence of SAM and increased resting LVOT gradient were associated with increased arrhythmia recurrence. Meanwhile white race, higher NYHA class, higher LVEF (HR 0.94, 95% CI 0.90–0.98), and higher LVOT gradient by Valsalva were associated with reduced arrhythmia recurrence. Comprehensive list of variables studied including univariable analyses are in Supplementary Tables 3–12.

**Safety outcomes**. Two (10%) DOF-AF patients had QT prolongation during inpatient loading leading to discontinuation, whereas the 3 (6%) STL-AF patients who had QT prolongation developed it at 1, 7, and 17 months after loading. One of 3 STL-AF patients who stopped the drug due to bradyarrhythmia had histories of alcohol septal ablation and primary prevention single

**Table 2 Baseline echocardiographic and electrocardiogram parameters prior to drug initiation.**

| | Dofetilide – AF (N = 21) | Sotalol – AF (N = 52) | Sotalol – VA (N = 18) | Sotalol – All (N = 61) | P-value (Dofetilide – AF vs Sotalol – AF) |
|---|---|---|---|---|---|
| LVEF (%) | 61.7 ± 7.5 | 63.4 ± 14.1 | 58.4 ± 15.0 | 63.4 ± 13.3 | 0.06 |
| LA diameter in parasternal long axis (cm) | 4.6 ± 1.0 | 5.3 ± 5.2 | 5.0 ± 1.0 | 5.2 ± 4.8 | 0.667 |
| LA volume index (ml/m$^2$) | 45.3 ± 16.6 | 44.4 ± 19.5 | 53.4 ± 23.4 | 44.4 ± 19.1 | 0.978 |
| IVSd (cm) | 1.4 ± 0.4 | 1.7 ± 0.6 | 1.7 ± 0.4 | 1.7 ± 0.6 | 0.007 |
| LPWd (cm) | 1.2 ± 0.3 | 1.3 ± 0.3 | 1.4 ± 0.2 | 1.3 ± 0.3 | 0.284 |
| Systolic Anterior Motion (N, % present) | 3 (14.3) | 11 (21.2) | 2 (11.1) | 12 (19.7) | 0.529 |
| Obstructive Gradient (rest or provoked, N, % present) | 3 (14.2) | 13 (25.0) | 4 (22.2) | 14 (23.0) | 0.364 |
| Resting Obstructive gradient (N, % present) | 2 (9.5) | 7 (13.4) | 0 (0.0) | 17 (11.5) | 0.668 |
| Provoked Obstructive gradient (by Valsalva or exercise, N, % present) | 2 (9.5) | 12 (23.1) | 4 (22.2) | 12 (19.7) | 0.332 |
| Resting LVOT gradient (mmHg) | 11.5 ± 18.4 | 19.8 ± 35.3 | 9.3 ± 8.2 | 18.5 ± 33.6 | 0.316 |
| Valsalva LVOT gradient (mmHg) | 19.2 ± 26.7 | 31.8 ± 49.3 | 21.1 ± 20.9 | 29.2 ± 46.6 | 0.297 |
| Exercise LVOT gradient (mmHg) | 9.3 ± 8.6 | 30.8 ± 32.0 | 51.6 ± 46.6 | 31.6 ± 29.9 | 0.178 |
| Baseline QTc (ms) | 439.5 ± 107.5 | 454.0 ± 35.5 | 455.7 ± 25.0 | 452.3 ± 33.9 | 0.583 |
| Baseline QRSd (ms) | 103.6 ± 33.9 | 114.2 ± 29.2 | 126.4 ± 25.9 | 114.7 ± 28.2 | 0.583 |

*LVEF left ventricular ejection fraction, LA left atrium, IVSd diastolic interventricular septal thickness, LPWd diastolic left ventricular posterior wall thickness, LVOT left ventricular outflow tract.*

chamber ICD, and developed complete heart block followed by non-sustained torsades de pointes (Fig. 2) in the setting of severe influenza pneumonia and subsequent acute kidney injury. Intravenous magnesium was given, STL was discontinued, his ICD's lower rate limit (previously VVI 40) was increased, and he underwent addition of a right atrial lead. No patients developed sustained torsades de pointes. HF hospitalization rates were similarly low in DOF-AF and STL-AF groups at 1 year (0 vs 3 patients, $p = 0.176$), as were CV death (0 vs 1 patient) and all-cause death (0 vs 2 patients). The single CV death was in a STL patient who suffered a cardiac arrested with subsequent cardiogenic shock due to an anterior ST-elevation myocardial infarction (no arrhythmias were recorded on patient's ICD). Multivariable predictors of drug discontinuation are reported in Supplementary Table 2.

## Discussion

This study is the largest cohort of HCM patients on DOF and STL to date, with longer duration of follow-up compared to the only 2 other studies in this field[14,15]. Our study is also the first to study safety and efficacy of STL for VA in this population, as both prior studies only studied HCM patients with AF. Both DOF and STL are Vaughan-Williams class III antiarrhythmic drugs that prolong repolarization by blocking potassium channels, increasing risk for QT prolongation and Torsades de Pointes especially in instances of impaired renal function. DOF is used primarily for atrial tachyarrhythmias (has not been extensively studied for VA), does not have significant hemodynamic effects, and should be crosschecked for potential drug-drug interactions that can increase DOF serum levels. STL has demonstrated efficacy in atrial tachyarrhythmias and both scar-based VT and premature ventricular contractions, and has additional beta-blocking effects, which are therapeutic for most obstructive HCM patients, but may be deleterious for end-stage non-obstructive HCM patients who have developed LV systolic dysfunction. Based on our experience and the data in this study, our contemporary strategy is to offer STL as a first-line anti-arrhythmic medication (along with catheter ablation if indicated/feasible using a shared decision-making approach) to HCM patients without significant renal dysfunction, LV systolic dysfunction or QT prolongation. We reserve DOF for patients who are STL-intolerant or who have specific drug-drug interactions precluding STL. While the data from our study and others is reassuring regarding low incidence of STL/DOF pro-arrhythmia in HCM[14,15], we still find it more reassuring to initiate these drugs in HCM patients with ICDs, and somewhat prioritizing catheter ablation for patients without ICDs. However, it should be noted that a little over half of our cohort on STL or DOF had ICDs and there was only one documented non-sustained pro-arrhythmia event, as described above. After loading, we routinely monitor with ECG and basic metabolic panel for electrolytes and renal function every 3 months on DOF and every 6 months on STL.

Our study suggests modest efficacy of DOF and STL for both AF and VA in HCM patients, with 40–50% recurrence at 1 year, with increasing recurrence out to a 7-year follow-up period (Fig. 1). While sobering, our study's AF recurrence rates are similar to those of AF catheter ablation for HCM patients, which range 52–71% after a single procedure, and improved to 34–61% allowing for multiple procedures, even at high-volume, experienced AF ablation centers[19–21]. These high recurrence rates are likely due to extensive atrial fibrosis and hypertrophy due to chronically increased left atrial pressure combined with direct atrial myocyte effects of HCM patients' sarcomere mutations. Accordingly, older studies of amiodarone for AF rhythm control in HCM demonstrated similar long-term AF recurrence rates of

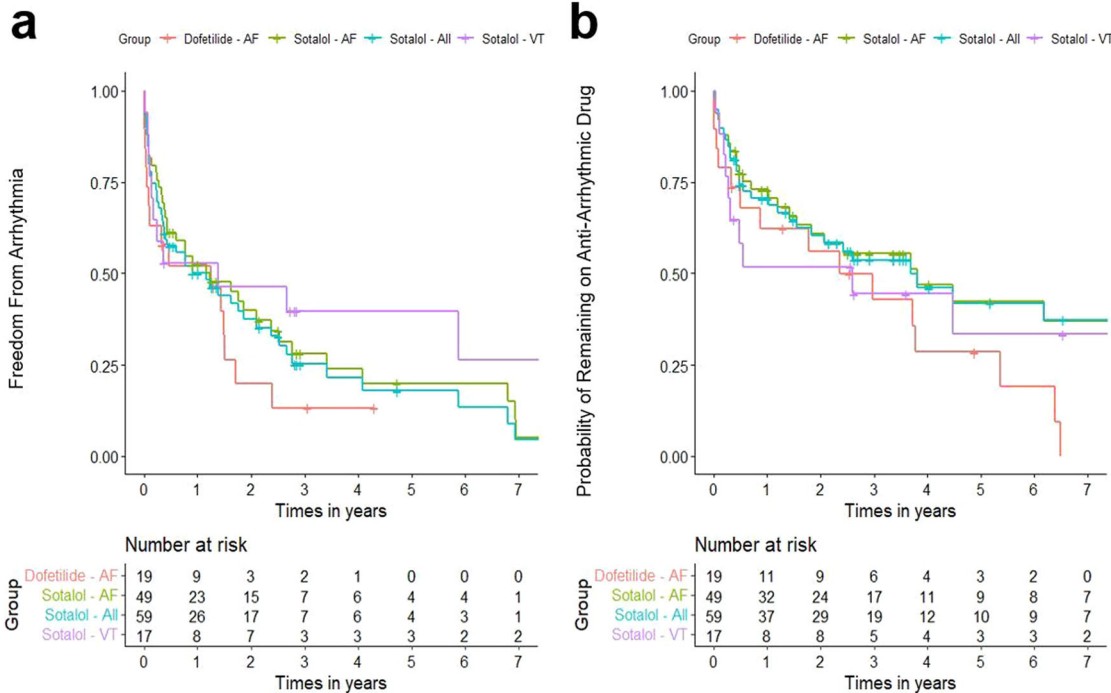

**Fig. 1 Kaplan–Meier curves of dofetilide (n = 21) and sotalol (n = 61) efficacy and safety in hypertrophic cardiomyopathy patients with atrial fibrillation and ventricular arrhythmias. a** Instances of arrhythmia recurrence over time across study groups (red = AF patients on dofetilide, green = AF patients on sotalol, blue = all patients on sotalol, purple = VA patients on sotalol). **b** Instances of drug discontinuation over time across study groups (red = AF patients on dofetilide, green = AF patients on sotalol, blue = all patients on sotalol, purple = VA patients on sotalol).

**Table 3 Short- and long-term safety and efficacy outcomes.**

|  | Dofetilide – AF (N = 21) | Sotalol – AF (N = 52) | Sotalol – VA (N = 18) | Sotalol – All (N = 61) | P-value (Dofetilide – AF vs Sotalol – AF) |
|---|---|---|---|---|---|
| Total follow-up days, median (IQR) | 2630 (1569, 4420) | 1272 (951, 3778) | 1385 (1004, 3986) | 1286 (953, 3155) | 0.072 |
| Total days on treatment, median (IQR) | 563 (96, 1362) | 566 (165, 1321) | 190 (102, 1227) | 616 (156, 1303) | 0.514 |
| HF hospitalizations at 1 year, N (%) | 0 (0.0) | 3 (5.8) | 1 (5.6) | 3 (4.9) | 0.176 |
| Cardiovascular death at 1 year, N (%) | 0 (0.0) | 1 (1.9) | 0 (0.0) | 1 (1.6) | 0.495 |
| All-cause death at 1 year, N (%) | 0 (0.0) | 2 (3.8) | 0 (0.0) | 2 (3.3) | 0.356 |
| Arrhythmia recurrence at 1 year, N (%) | 9 (42.9) | 23 (44.2) | 9 (50.0) | 31 (50.8) | 0.665 |
| Long-term arrhythmia recurrence, N (%) | 15 (71.4) | 38 (73.1) | 13 (72.2) | 51 (83.6) | 0.223 |
| Discontinued at 1 year, N (%) | 8 (38.1) | 15 (29.4) | 8 (44.4) | 18 (29.5) | 0.358 |
| Long-term discontinuation, N (%) | 16 (76.2) | 32 (61.5) | 12 (66.7) | 36 (59.0) | 0.084 |
| Reason for discontinuation, N (%) |  |  |  |  |  |
| Inefficacy | 6 (28.9) | 14 (26.9) | 3 (16.7) | 17 (27.9) |  |
| Adverse event | 2 (9.5) | 11 (21.2) | 4 (22.2) | 11 (18.0) |  |
| HF signs and symptoms | 0 (0.0) | 4 (7.7) | 1 (5.6) | 4 (6.6) |  |
| QT prolongation | 2 (9.5) | 3 (5.8) | 1 (5.6) | 3 (4.9) |  |
| Hypotension | 0 (0.0) | 1 (1.9) | 1 (5.6) | 1 (1.6) |  |
| Bradyarrhythmia | 0 (0.0) | 3 (5.8) | 1 (5.6) | 3 (4.9) |  |
| Side-effects | 3 (14.3) | 4 (7.7) | 5 (27.8) | 8 (13.1) |  |
| Fatigue, dyspnea, or weakness | 1 (4.8) | 2 (3.8) | 3 (16.7) | 4 (6.6) |  |
| Dizziness | 0 (0.0) | 1 (1.9) | 2 (11.1) | 3 (4.9) |  |
| GI intolerance | 0 (0.0) | 1 (1.9) | 0 (0.0) | 1 (1.6) |  |
| Depression | 1 (4.8) | 0 (0.0) | 0 (0.0) | 0 (0.0) |  |
| Hives | 1 (4.8) | 0 (0.0) | 0 (0.0) | 0 (0.0) |  |
| Other (Heart transplant, patient preference, guideline non-compliance due to LVH) | 3 (14.3) | 1 (1.9) | 0 (0.0) | 1 (1.6) |  |

*IQR* inter-quartile range, *HF* heart failure, *LVH* left ventricular hypertrophy.

33%[22] and 55%[16]. Our cohort's relatively low loading doses of STL (Supplementary Table 1) may also have limited efficacy.

We meticulously studied our cohort for STL and DOF safety concerns, including QTc prolongation, HF (for STL), and sudden death. A majority of our cohort had IVSd > 1.5 cm (above which the 2014 and 2019 AF guidelines advise against use of STL or DOF; mean IVSd 1.7 ± 0.6 cm), so our results were overall quite reassuring. QTc prolongation precluded DOF loading in only

10% of our patients, and contributed to post-loading STL discontinuation in 6%, comparable to prior studies[14]. Only 5% of our STL patients had a HF hospitalization over our median 4.4-year follow-up (comparable to HF incidence in a broad HCM population over such a period[23] and not clearly linked to STL's negative inotropy). Although some patients did have STL discontinued due to HF concern, the overall low HF hospitalization rates might suggest that STL's negative inotropic effect has some benefit in the obstructive HCM population. Rates of death were reassuringly low across all of our study groups.

Our study has several limitations. Despite our study having a larger cohort than the only two prior studies on this topic[14,15], it is still a small, single-center study. We did not employ a standardized approach to anti-arrhythmic drug selection or to STL dosing and loading (much of which was done as outpatient; our contemporary practice is to recommend inpatient loading for both STL and DOF, but to allow cautious outpatient STL loading with close electrolyte, renal function and QT monitoring for patients who have ICDs), with only a little over half of our STL patients maintained on a dose of 120 mg BID or higher (Supplementary Table 1). More aggressive STL dosing may have improved efficacy, but compromised safety. Only 13% of our patients had resting obstructive LVOT gradients (23% inducible LVOT obstruction), which is lower than the convention that 2/3 of HCM patients have obstructive physiology, although 28% of

our patients had undergone effective septal reduction therapy (21% myectomy and 7% alcohol septal ablation), and were only non-obstructive at the time of drug initiation. Our study also lacked a drug-free (i.e., rate control) control group or comparator groups of patients on amiodarone (for AF or VA) or disopyramide (for AF). We avoided these control groups as rate control is very rarely effective in HCM patients who tolerate AF poorly, and an amiodarone control group would be very challenging to propensity-score match in a study of HCM patients who trend quite young and for whom the risks of long-term toxicities are intolerable. Disopyramide is predominantly used for LVOT obstruction in our institution, which would have introduced bias in its use as a control group. Thus, we lacked a sufficient number of patients on any of these 3 strategies (with balanced baseline characteristics to our STL and DOF groups) to subsequently propensity-score match control groups. We chose AF recurrence (binary) as an outcome which, admittedly, is inferior to quantitative AF burden (which more clearly correlates with quality of life). Unfortunately, AF burden was too inconsistently quantified in this historical cohort study (some of which pre-dates the use of longer-term ambulatory ECG monitors) to further analyze.

Finally, our multivariable analysis for predictors of arrhythmia recurrence included both AF and VA, acknowledging the mechanisms of each is different. We did further sub-analyze for predictors of recurrence of AF and VA separately (Supplemental Table 7), but univariable analyses did not yield any significant predictors, so multivariable analysis was not performed. Perplexing was that the presence of provoked LVOT gradients that was associated with decreased arrhythmia events; meanwhile, a SAM hazard ratio of 8.89 for arrhythmia recurrence seemed out of proportion to the weaker hazard ratios between LVOT gradients and arrhythmia recurrence. It is understood that HCM patients with obstruction, of which SAM is an accepted mechanism, will run higher risks of reduced cardiac output, myocardial ischemia, interstitial fibrosis, HF, and arrhythmias[12,24–27]. However, our data collection of significant gradients found only 5 patients at rest and 10 patients with provocation (3 with myectomy). Thus, even a multivariable model likely could not overcome confounding and incorrectly suggested provoked gradients as "protective" against arrhythmia recurrence. Only 12 patients in our entire cohort had documented SAM. Overall, the multivariable analysis findings must be viewed as hypothesis generating at best with discrepancies likely explained by incomplete or missing data leading to smaller

**Table 4 Multivariate Cox proportional hazard models of predictors for arrhythmia recurrence.**

|  | HR | 95% CI | P-value |
|---|---|---|---|
| Age | 1.09 | 1.03–1.17 | 0.007 |
| Race (white) | 0.12 | 0.03–0.47 | 0.003 |
| NYHA Class | 0.20 | 0.07–0.62 | 0.005 |
| LVEF | 0.94 | 0.90–0.98 | 0.007 |
| IVSd | 2.96 | 1.01–8.69 | 0.048 |
| Systolic Anterior Motion | 8.89 | 1.91–41.34 | 0.005 |
| Resting LVOT gradient | 1.15 | 1.04–1.27 | 0.006 |
| Valsalva LVOT gradient | 0.95 | 0.92–0.99 | 0.015 |

Wald test p = 0.004.
NYHA New York Heart Association, LVEF left ventricular ejection fraction, IVSd diastolic interventricular septal thickness, LVOT left ventricular outflow tract gradient.

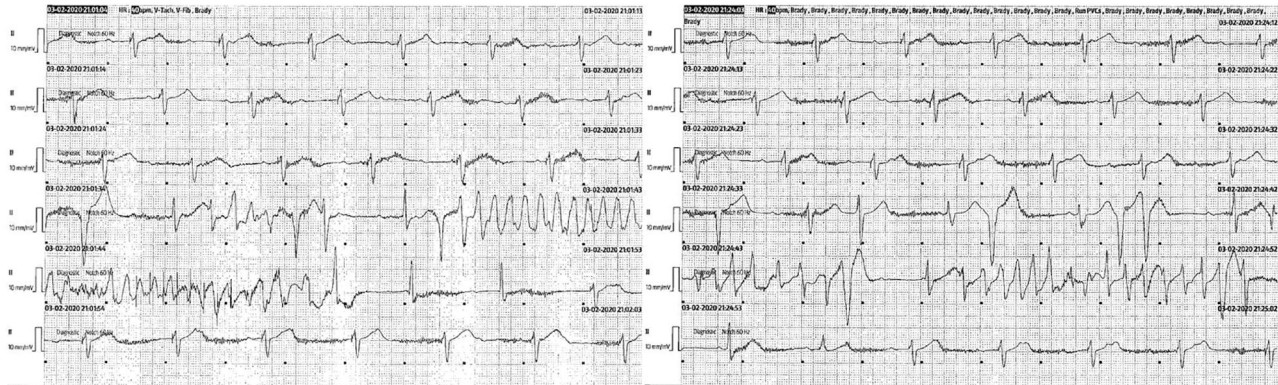

**Fig. 2 A case of hypertrophic cardiomyopathy in a patient on sotalol with torsades de pointes in the setting of influenza pneumonia and acute kidney injury.** Telemetry strips of a patient on sotalol for atrial fibrillation with history of alcohol septal ablation and primary prevention single chamber ICD who presented with complete heart block and bradycardia-induced torsades de pointes in the setting of severe influenza pneumonia with concomitant acute kidney injury. No defibrillations were required. Sotalol was discontinued without improvement, his lower rate limit was increased, and the patient eventually underwent upgrade to a dual-chamber ICD.

sample sizes and an underpowered study not infrequently encountered for retrospective studies of this unique population.

## Conclusions

Our study supports satisfactory safety but only modest efficacy (comparable to that of catheter ablation) of DOF and STL for the treatment of AF and VA in HCM patients, and supports the 2020 HCM guidelines' recommendation that use of these agents is "reasonable".

## Data availability

Data sets were generated as a .csv file from extracted patient information from the electronic medical record system of our hospital institution. They are currently stored within a secure Microsoft OneDrive storage system owned by our academic hospital institution, and are available from the corresponding authors upon request. Source data behind Fig. 1 specifically can be viewed in Supplementary Data 1.

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

## Author contributions

C.C., M.L., and B.N. designed the study and acquired funding. C.C. and M.L. collected the data. C.C. and B.N. primarily wrote and reviewed the manuscript. Y.B. performed statistical analysis of the data and wrote the statistical portions of the manuscript. H.C., E.S., and A.M. reviewed the study and manuscript and provided feedback for revisions.

## Competing interests

The authors declare no competing interests.
