## [Peer Review File · Communications Medicine]

Reviewers' comments:

Reviewer #1 (Remarks to the Author):

The study by Chen et al "Efficacy and Safety of Dofetilide and Sotalol in Patients with Hypertrophic Cardiomyopathy" describes use of sotalol and dofetilide for arrhythmia suppression in patients with hypertrophic cardiomyopathy (HCM). This is a particularly relevant issue as there is conflicting recommendations between societal guidelines as to safety of these drugs, and not infrequently, these AADs are withheld from young HCM patients given concerns that these drugs could be facilitate ventricular tachyarrhythmias (proarrhythmic) and with patients instead started on amiodarone which has potential long term toxicities. At present data on safety of these drugs is limited to very small case series. Thereby, this additional data regarding safety and efficacy is welcome and important addition to the HCM literature.

Major suggestions:

1- It appears that sotalol use for both ventricular arrhythmias as well as atrial fibrillation is part of the multivariable analysis as predictors for arrhythmia recurrence. However, the mechanism for recurrent ventricular arrhythmias is likely markedly different for mechanism of recurrent AF. For this reason, I would recommend performing this analysis separately for AF in order to provide a more clinically relevant analysis.

2- Regarding patient selection between sotalol or dofetilide for AF treatment. I understand the authors comment that it is challenging to retrospectively describe the decision making in choosing between these drugs on individual patient level. However, it would be helpful for the authors to describe the major factors that generally goes into their decision making between these two drugs. This is crucial information that would benefit the practicing cardiology community who has less experience in hypertrophic cardiomyopathy in order to help guide their decision between these drugs. This could be included either in methods or in discussion.

3- It remains unclear the subset of patients initiated on sotalol for ventricular arrhythmias. What was HCM phenotype for these patients? How many had LV apical aneurysms and endstage HCM with systolic dysfunction(phenotypes associated with high rates of recurrent VT/VF)? Is there LGE extent data available for this group? This would help place efficacy data in better context.

Minor comments:

1- Of patients with recurrent AF, how many developed PAF versus persistent AF? How many patients with recurrent AF/treatment failures went on to have AF ablations? While recurrence rate is high, given

retrospective nature of analysis, unable to comment on AF burden, which is likely substantially decreased with these AAD given the high rates of permanent AF at study entry. There is limitations in using AF recurrence as an endpoint as does not take into account potential to decrease AF burden and improve quality of life.

2- Of the patients initiated on both sotalol and dofetilide, which was initiated first? What was recurrent rate on second drug in this subset of patients? I would anticipate it to be higher if initial discontinuation was due to inefficacy.

3- Data on IVSD difference between sotalol-AF and dofetilide AF appears twice in results section (lines 157 and 165).

4- "trend at long term follow up towards less frequent discontinuation in the STL-AF group ($p=0.223$)". This is a non-significant p-value that does not appear close to approaching clinical significance. Would change this sentence to state there is no difference between discontinuation rates between the drugs.

5- While there is concern in general about negative inotropic effect of sotalol in other cardiovascular disease, this could be beneficial in obstructive HCM and potentially even in non-obstructive HCM. This likely explains why heart failure hospitalization rate is so low in this group.

Reviewer #2 (Remarks to the Author):

The authors present a study titled Efficacy and Safety of Dofetilide and Sotalol in Patients With Hypertrophic Cardiomyopathy. In general, the data about these medications in HCM is scarce. Although the manuscript could be of interest to HCM readers, it cannot be recommended for publication as it stands. Please find some of my comments below.

It is a retrospective study based on "search criteria" from the medical record with many limitations and selection bias. These criteria have to be described in the Methods section.

Was sotalol/dofetilide treatment started in inpatients or outpatients? Please update the Methods section regarding drug initiation and subsequent monitoring/follow-up.

Safety, line 191 - what does " well after the loading period" mean? Please present accurate data. More data about dosing and loading should be presented.

As the authors pointed out, the study lacked the comparison with other antiarrhythmics, e.g. amiodarone or disopyramide. Why? What was the reason for choosing these antiarrhythmics? Amiodarone/disopyramide side effects?

I'm afraid I have to disagree with the conclusion. The efficacy is unsatisfactory, and the conclusion sentences should be modified in line with the presented data.

The authors should discuss why they consider randomized, prospective studies comparing these agents unlikely in the Discussion section. It does not belong to the Conclusion section.

Table 4 - the authors should discuss why there is a difference in MV analysis with resting and provoked pressure gradients. Similarly, the SAM and the HR of arrhythmia recurrence. Please define SAM and try to explain. Is this plausible? Why were those chosen as variables to MV analysis? Please consider the number of events and the number of variables. What is the power of this analysis?

Patients with sotalol for nonsustained ventricular tachycardia or premature ventricular complexes were excluded. Please explain.

Response to Reviewers

Thank you for your careful review of our manuscript. We feel that our manuscript has been markedly improved by your suggestions. We detail our responses below.

Reviewer #1

1. It appears that sotalol use for both ventricular arrhythmias as well as atrial fibrillation is part of the multivariable analysis as predictors for arrhythmia recurrence. However, the mechanism for recurrent ventricular arrhythmias is likely markedly different for mechanism of recurrent AF. For this reason, I would recommend performing this analysis separately for AF in order to provide a more clinically relevant analysis.

Thank you for your review of the manuscript and this important distinction. We performed this analysis specifically as univariable analysis for AF (only) recurrence on sotalol in our Supplemental Tables file (page 7 of the document under “Supplemental Table 7”). As there were only 2 variables that met statistical significance (race and exercise LVOT gradient), we did not perform a multivariable analysis. To your much-appreciated credit, we have added this into our limitations of the manuscript (page 12):

“Finally, our multivariable analysis for predictors of arrhythmia recurrence included both AF and VA, acknowledging the mechanisms of each is different. We did further sub-analyze for predictors of recurrence of AF and VA separately (Supp. Table 7), but univariate analyses did not yield any significant predictors, so multivariable analysis was not performed.”

2. Regarding patient selection between sotalol or dofetilide for AF treatment. I understand the authors comment that it is challenging to retrospectively describe the decision making in choosing between these drugs on individual patient level. However, it would be helpful for the authors to describe the major factors that generally goes into their decision making between these two drugs. This is crucial information that would benefit the practicing cardiology community who has less experience in hypertrophic cardiomyopathy in order to help guide their decision between these drugs. This could be included either in methods or in discussion.

We thank you for bringing up this suggestion and have added the following into our Discussion section (page 9). We hope it helps guide readers:

“Both DOF and STL are Vaughan-Williams class III antiarrhythmic drugs that prolong repolarization by blocking potassium channels, increasing risk for QT prolongation and Torsades de Pointes especially in instances of impaired renal function. DOF is used primarily for atrial tachyarrhythmias (has not been extensively studied for VA), does not have significant hemodynamic effects, and should be crosschecked for potential drug-drug interactions that can increase DOF serum levels. STL has demonstrated efficacy in atrial tachyarrhythmias and both scar-based VT and premature ventricular contractions, and has additional beta-blocking effects which are therapeutic for most obstructive HCM patients, but may be deleterious for end-stage

non-obstructive HCM patients who have developed LV systolic dysfunction. Based on our experience and the data in this study, our contemporary strategy is to offer STL as a first-line anti-arrhythmic medication (along with catheter ablation if indicated/feasible using a shared decision-making approach) to HCM patients without significant renal dysfunction, LV systolic dysfunction or QT prolongation. We reserve DOF for patients who are STL-intolerant or who have specific drug-drug interactions precluding STL. While the data from our study and others is reassuring regarding low incidence of STL/DOF pro-arrhythmia in HCM^{14,15}, we still find it more reassuring to initiate these drugs in HCM patients with ICDs, and somewhat prioritizing catheter ablation (or amiodarone) for patients without ICDs. However, it should be noted that a little over half of our cohort on STL or DOF had ICDs and there was only one documented non-sustained pro-arrhythmia event, as described above. After loading, we routinely monitor with ECG and basic metabolic panel for electrolytes and renal function every 3 months on DOF and every 6 months on STL.”

3. It remains unclear the subset of patients initiated on sotalol for ventricular arrhythmias. What was HCM phenotype for these patients? How many had LV apical aneurysms and endstage HCM with systolic dysfunction(phenotypes associated with high rates of recurrent VT/VF)? Is there LGE extent data available for this group? This would help place efficacy data in better context.

Thank you for bringing up these vital points. We experienced the challenges often faced in retrospective analyses with missing information/poor documentation. For instance, of the 18 patients who were on STL for VA, only 8 (44.4%) had CMRs (which 5 reports did not quantify LGE extent). Three of the 18 had LV aneurysms described but with normal LVEF at time of STL initiation. Finally, 3 of 18 (16.7%) had an LVEF < 35% at time of drug initiation (none of these had LV aneurysms noted). These would suggest in our cohort STL was not initiated more likely due to more severe phenotypes of HCM, but with the caveat of incomplete data. We have added this to the Results portion of the manuscript (page 8):

“Within the STL VA population, only 8 (44%) patients had CMR of which 5 of these did not quantify late gadolinium enhancement (LGE) extent. In this same group, 3 (17%) had LV aneurysms noted, and 3 (17%) had LVEF <50% at time of drug initiation.”

4. Of patients with recurrent AF, how many developed PAF versus persistent AF? How many patients with recurrent AF/treatment failures went on to have AF ablations? While recurrence rate is high, given retrospective nature of analysis, unable to comment on AF burden, which is likely substantially decreased with these AAD given the high rates of permanent AF at study entry. There is limitations in using AF recurrence as an endpoint as does not take into account potential to decrease AF burden and improve quality of life.

Thank you greatly for bringing up this important point regarding meaningful AF outcomes. In Table 1 (page 13) of the manuscript one can see that the baseline AF burdens at time of drug initiation were largely Persistent AF at about 95% for DOF and just under 70% for STL. There was only 1 patient in the entire cohort that was found to be in permanent AF at time of STL initiation (which was actually initiated for VA specifically).

We certainly agree looking at the resulting reduction of AF burden as a measure of improving quality of life is another way to assess drug efficacy. There are many challenges with retrospectively assessing AF burden as well. We were at the mercy of the quantity/quality and how specific to details were the follow-up visit data (did they have follow-up ECG/Holter/Zio evidence, how did they document, etc). Decisions have to be made at what duration of follow-up does one decide to reassess burden (for example, will there be more or less data available at 1 year vs 5 years?). Many patients have mixed picture AF with both paroxysmal and persistent burden, which further grays the distinction. Some of the AF ablations did not occur simply due to drug failure, but were already planned as longer-term substrate modification strategy (with the patient in sinus rhythm on DOF or STL at time of procedure) and with plans to remove DOF/STL post ablation. We have included this important topic in our limitations of the manuscript now (page 12):

“We chose AF recurrence (binary) as an outcome which, admittedly, is inferior to quantitative AF burden (which more clearly correlates with quality of life). Unfortunately, AF burden was too inconsistently quantified in this historical cohort study (some of which pre-dates the use of longer-term ambulatory ECG monitors) to further analyze.”

5. Of the patients initiated on both sotalol and dofetilide, which was initiated first? What was recurrent rate on second drug in this subset of patients? I would anticipate it to be higher if initial discontinuation was due to inefficacy.

DOF initiated first in 4 patients later switched to STL of which only 1 had documented AF recurrence at 1 year. STL initiated first in 6 patients later switched to DOF of which 2 had documented AF recurrence at 1 year. Vast majority of these patients in either group by time of chart review had AF recurrence eventually.

6. Data on IVSD difference between sotalol-AF and dofetilide AF appears twice in results section (lines 157 and 165) “trend at long term follow up towards less frequent discontinuation in the STL-AF group (p=0.223)”. This is a non-significant p-value that does not appear close to approaching clinical significance. Would change this sentence to state there is no difference between discontinuation rates between the drugs

Thank you for catching this error and for this feedback. We have deleted one of the duplicates have changed the wording in the manuscript as such (page 8):

“At 1 year, drug was discontinued in 8 (38%) DOF-AF and 15 (29%, p=0.358) STL-AF patients, respectively, with no difference in discontinuation rates between groups (p=0.223; Figure 1).”

7. While there is concern in general about negative inotropic effect of sotalol in other cardiovascular disease, this could be beneficial in obstructive HCM and potentially even in non-obstructive HCM. This likely explains why heart failure hospitalization rate is so low in this group.

Thank you for bringing up this point. We have included it in our discussion (page 10):

“Only 5% of our STL patients had a HF hospitalization over our median 4.4 year follow-up (comparable to HF incidence in a broad HCM population over such a period²¹ and not clearly linked to STL’s negative inotropy). Although some patients did have STL discontinued due to HF concern, the overall low HF hospitalization rates might suggest that STL’s negative inotropic effect has some benefit in the obstructive HCM population.”

Reviewer #2

1. It is a retrospective study based on "search criteria" from the medical record with many limitations and selection bias. These criteria have to be described in the Methods section.

We thank you for instructing us on this detail. We have added further explanation to our manuscript under the Methods section (page 5):

“We searched our institution’s electronic medical record for patients with a diagnosis of “hypertrophic cardiomyopathy” (obstructive or non-obstructive including all relevant ICD-9 and ICD-10 codes within problem list and past medical history), and then filtered them by searching for “dofetilide” or “sotalol” on current or prior medication lists. Four patient groups were collected according to arrhythmia and medication: DOF-AF, STL-AF, STL-VA, and STL-All (both AF and VA). Manual chart review confirmed correct diagnoses of HCM not just by written documentation but also corroborated by echocardiography or cardiac MRI. Patients were excluded if found with erroneous or mislabeled diagnoses, age less than 18 years at time of drug initiation, and for lack of documentation regarding drug loading, arrhythmia recurrences, or follow-up.”

2. Was sotalol/dofetilide treatment started in inpatients or outpatients? Please update the Methods section regarding drug initiation and subsequent monitoring/follow-up.

Thank you for bringing up this point. Given the retrospective nature of our analysis, the patients had already been initiated either inpatient or outpatient (and thus not from our own planned decision). We found that all 21 patients on DOF were loaded inpatient, 32 (61%) patients on STL for AF were loaded inpatient, and 11 (61%) patients on STL for VA were loaded inpatient – and we felt it more appropriate to place this in the results section (found on bottom of page 7):

“All 21 patients on DOF were loaded inpatient while 32 (62%) STL-AF and 11 (61%) STL-VA patients were loaded inpatient.”

We also describe our contemporary loading strategy in the Discussion section (page 11):

“...our contemporary practice is to recommend inpatient loading for both STL and DOF, but to allow cautious outpatient STL loading with close electrolyte, renal function and QT monitoring for patients who have ICDs”

And for monitoring (page 10):

“After loading, we routinely monitor with ECG and basic metabolic panel for electrolytes and renal function every 3 months on DOF and every 6 months on STL.”

3. Safety, line 191 - what does " well after the loading period" mean? Please present accurate data. More data about dosing and loading should be presented.

We apologize for the lack of clarity. Our meaning was that the QT prolongation happened after the routine 3-day period of monitoring the patient's QT during inpatient initiation. Specifically, in the 3 STL AF patients who had QT prolongation requiring discontinuation this occurred at 1 month and 7 days, 1 year and 5 months, and 7 months respectively from the time of STL initiation. We have clarified the manuscript to read as such (page 9):

“Two (10%) DOF-AF patients had QT prolongation during inpatient loading leading to discontinuation, whereas the 3 (6%) STL-AF patients who had QT prolongation developed it at 1, 7 and 17 months after loading.”

4. As the authors pointed out, the study lacked the comparison with other antiarrhythmics, e.g. amiodarone or disopyramide. Why? What was the reason for choosing these antiarrhythmics? Amiodarone/disopyramide side effects?

We appreciate this comment/question. At our institution, we use disopyramide very rarely for AF rhythm control, and mainly use (used) it for relief of LVOT obstruction (in the pre-mavacamten era). Similarly, we avoid amiodarone except in the older, more end-stage HCM population. Similarly, nearly all of our HCM patients with AF have AF-attributable symptoms and are rarely on rate control. Thus, none of the 3 control groups of amiodarone, rate control, or disopyramide were feasible or practical. Based on your input, we further describe this in the Limitations section of our Discussion (pages 12):

“Our study also lacked a drug-free (i.e. rate control) control group or comparator groups of patients on amiodarone (for AF or VA) or disopyramide (for AF). We avoided these control groups as rate control is very rarely effective in HCM patients who tolerate AF poorly, and an amiodarone control group would be very challenging to propensity-score match in a study of HCM patients who trend quite young and for whom the risks of long-term toxicities are intolerable. Disopyramide is predominantly used for LVOT obstruction in our institution, which would have introduced bias in its use as a control group. Thus, we lacked a sufficient number of patients on any of these 3 strategies (with balanced baseline characteristics to our STL and DOF groups) to subsequently propensity-score match control groups.”

5. I'm afraid I have to disagree with the conclusion. The efficacy is unsatisfactory, and the conclusion sentences should be modified in line with the presented data.

We thank you for this input. Compared to rhythm control efficacy for patients without structural heart disease, our recurrence rates are indeed high. But HCM is an aggressive disease that likely affects the atrium both via chronically elevated filling pressures as well as direct atrial myocyte/sarcomeric derangement by the pathogenic variant. Thus, even at a high volume, experienced AF ablation centers (Cleveland Clinic, high-volume European centers), recurrence

rates are comparable to ours. For example, in catheter ablation of AF in HCM patients, recurrence rates have been seen to go from ~42% at 6 months to ~57% at 1 year (Bassiouny M, Lindsay BD, Lever H, et al. Outcomes of nonpharmacologic treatment of atrial fibrillation in patients with hypertrophic cardiomyopathy. *Heart Rhythm*. 2015;12(7):1438-1447. doi:10.1016/j.hrthm.2015.03.042). We have added two more references re: AF ablation in HCM with similar recurrence rates. Finally, we found some older studies of amiodarone for AF rhythm control in HCM (admittedly from the 1980s and 1990s) which demonstrated recurrence rates of 33-55%. We have updated the Limitations section of our Discussion (pages 10-11):

“While sobering, our study’s AF recurrence rates are similar to those of AF catheter ablation for HCM patients, which range 52-71% after a single procedure, and improved to 34-61% allowing for multiple procedures, even at high-volume, experienced AF ablation centers¹⁹⁻²¹. These high recurrence rates are likely due to extensive atrial fibrosis and hypertrophy due to chronically increased left atrial pressure combined with direct atrial myocyte effects of HCM patients’ sarcomere mutations. Accordingly, older studies of amiodarone for AF rhythm control in HCM demonstrated similar long-term AF recurrence rates of 33%²² and 55%¹⁶.”

We feel the word “satisfactory” in our Conclusion is sufficiently modest regarding efficacy, but can further temper it if the reviewer insists.

6. The authors should discuss why they consider randomized, prospective studies comparing these agents unlikely in the Discussion section. It does not belong to the Conclusion section.

Thank you for providing this instruction. Randomized, prospective studies comparing these agents are unlikely because they are long-established antiarrhythmic drugs used in the general population with understood efficacy/safety profile, have received some acceptance in the ACC/AHA HCM guidelines of 2020 (although not the AF guidelines of 2014), and such a trial which would require long-term follow-up would be challenging to fund. We have rephrased this sentence in the Conclusion section to the following (page 12):

“Without a current randomized, prospective study of these agents, confirmatory data from larger cohorts derived from multi-center HCM registries would further bolster HCM providers’ confidence in the use of these agents.”

7. Table 4 - the authors should discuss why there is a difference in MV analysis with resting and provoked pressure gradients. Similarly, the SAM and the HR of arrhythmia recurrence. Please define SAM and try to explain. Is this plausible? Why were those chosen as variables to MV analysis? Please consider the number of events and the number of variables. What is the power of this analysis?

Thank you for bringing up these responses. We found these findings somewhat perplexing and challenging to think through in looking for plausible explanation. We assume that patients with higher degrees of obstruction in HCM will run the higher risk of reduction in cardiac output, myocardial ischemia, and interstitial fibrosis which can predispose to both AF and VA’s. Thus patients with higher resting pressure gradients already are at high baseline risk for these processes and thus appear to be predictors for increased arrhythmia events. We do not have a

clear explanation why the presence of provoked pressure gradients seemed to be associated with decreased arrhythmia events, and it may be due to overall low number of patients that had recorded provokable gradients (only 11 patients in the entire cohort). Similarly, the very high HR for SAM with the wide range of CI 1.91-41.34 is also likely explained by small sample size (only 12 had documented SAM). Similarly, SAM or the systolic anterior motion of the mitral valve is the result of abnormal flow vectors caused by a combination of septal hypertrophy, long leaflets, and anterior displacement of the papillary muscles and mitral valve apparatus that lead to worsening LVOT obstruction which again as mentioned can lead to increased risk for arrhythmia events. Overall, we are most likely underpowered and the small sample size led to these findings.

8. Patients with sotalol for nonsustained ventricular tachycardia or premature ventricular complexes were excluded. Please explain.

Thank you for highlighting this point. We excluded these conditions because unlike AF and sustained VT/VF/ICD therapies, “recurrence” would be hard to adjudicate in a historical cohort. Furthermore, the number of patients for which sotalol was used for NSVT or PVCs was quite small, and we did not want to make our STL-VA group more heterogeneous.

Reviewers' comments:

Reviewer #1 (Remarks to the Author):

I appreciate the authors response and substantial changes to the manuscript based on my queries.

I have one minor residual comments:

The new final sentence to conclusion is unclear ("Without a current randomized, prospective study of these agents, confirmatory data

from larger cohorts derived from multi-center HCM registries would further bolster HCM providers' confidence in the use of these agents.)

I think would be reasonable to delete this sentence all-together or this sentence should be modified to improve clarity.

Reviewer #2 (Remarks to the Author):

The authors present a revised manuscript of the study named Efficacy and Safety of Dofetilide and Sotalol in Patients With Hypertrophic Cardiomyopathy. In their rebuttal letter, the Authors responded to most of my comments. Nevertheless, some of my concerns remain to be addressed with proper changes in the manuscript text.

Repeating my previous points:

Please further temper the word "satisfactory" in the Conclusion, as suggested.

Table 4 - the authors should discuss why there is a difference in MV analysis with resting and provoked pressure gradients. Similarly, the SAM and the HR of arrhythmia recurrence. Please define SAM and try to explain. Is this plausible? Why were those chosen as variables to

MV analysis? Please consider the number of events and the number of variables, i.e., the power of this analysis.

The Authors discussed some points in their response. Nevertheless, they did not explain how they chose the "perplexing" variables, nor did they make the changes in the manuscript text. This response and all the information discussed in the rebuttal letter must be embedded into the manuscript text - Discussion section.

Second Response to Reviewers

Thank you again for your careful review of our manuscript. We feel that our manuscript has been markedly improved by your suggestions. We detail our responses below.

Reviewer #1

1. I appreciate the authors response and substantial changes to the manuscript based on my queries.

I have one minor residual comments:

The new final sentence to conclusion is unclear ("Without a current randomized, prospective study of these agents, confirmatory data from larger cohorts derived from multi-center HCM registries would further bolster HCM providers' confidence in the use of these agents.")

I think would be reasonable to delete this sentence all-together or this sentence should be modified to improve clarity.

Thank you for your feedback regarding this concluding sentence. We felt that randomized, prospective studies comparing these agents are unlikely because they are long-established antiarrhythmic drugs used in the general population with understood efficacy/safety profile, have received some acceptance in the ACC/AHA HCM guidelines of 2020 (although not the AF guidelines of 2014), and such a trial which would require long-term follow-up would be challenging to fund. As such, the next best step is to gather larger data sets across multiple centers demonstrating safety and reasonable efficacy, which would bolster general confidence in using sotalol and dofetilide in HCM arrhythmia management.

As per your suggestion, we have deleted the final sentence from the conclusion.

Reviewer #2

1. The authors present a revised manuscript of the study named Efficacy and Safety of Dofetilide and Sotalol in Patients With Hypertrophic Cardiomyopathy. In their rebuttal letter, the Authors responded to most of my comments. Nevertheless, some of my concerns remain to be addressed with proper changes in the manuscript text.

Repeating my previous points:

Please further temper the word "satisfactory" in the Conclusion, as suggested.

Table 4 - the authors should discuss why there is a difference in MV analysis with resting and provoked pressure gradients. Similarly, the SAM and the HR of arrhythmia recurrence. Please define SAM and try to explain. Is this plausible? Why were those chosen as variables to MV analysis? Please consider the number of events and the number of variables, i.e., the

power of this analysis.

The Authors discussed some points in their response. Nevertheless, they did not explain how they chose the "perplexing" variables, nor did they make the changes in the manuscript text. This response and all the information discussed in the rebuttal letter must be embedded into the manuscript text - Discussion section.

We are truly grateful of your efforts to sharpen our paper to be more clear and effective.

As you have suggested, we have further tempered the conclusion sentence now to read:

Our study supports satisfactory safety but only modest efficacy (comparable to that of catheter ablation) of DOF and STL for the treatment of AF and VA in HCM patients, and supports the 2020 HCM guidelines' recommendation that use of these agents is "reasonable."

We have further clarified how SAM was defined in the Methods section (page 5):

"Systolic anterior motion (SAM) of the anterior mitral valve leaflet was defined based on mentioning in the echo report from qualitative assessment by the echo reader and not re-adjudicated by us.

We have modified our Discussion section (pages 12-13) now to read as follows:

"Finally, our multivariable analysis for predictors of arrhythmia recurrence included both AF and VA, acknowledging the mechanisms of each is different. We did further sub-analyze for predictors of recurrence of AF and VA separately (Supplemental Table 7), but univariable analyses did not yield any significant predictors, so multivariable analysis was not performed. Perplexing was that the presence of provoked LVOT gradients that was associated with decreased arrhythmia events; meanwhile, a SAM hazard ratio of 8.89 for arrhythmia recurrence seemed out of proportion to the weaker hazard ratios between LVOT gradients and arrhythmia recurrence. It is understood that HCM patients with obstruction, of which SAM is an accepted mechanism, will run higher risks of reduced cardiac output, myocardial ischemia, interstitial fibrosis, HF, and arrhythmias^{12,24-27}. However, our data collection of significant gradients found only 5 patients at rest and 10 patients with provocation (3 with myectomy). Thus, even a multivariable model likely could not overcome confounding and incorrectly suggested provoked gradients as "protective" against arrhythmia recurrence. Only 12 patients in our entire cohort had documented SAM. Overall, the multivariable analysis findings must be viewed as hypothesis-generating at best with discrepancies likely explained by incomplete or missing data leading to smaller sample sizes and an underpowered study not infrequently encountered for retrospective studies of this unique population."

If you find overall our attempt to explain these findings still too unsatisfactory we would also consider removing the multivariable model entirely from the paper – only with your approval.

REVIEWERS' COMMENTS:

Reviewer #2 (Remarks to the Author):

I appreciate the authors' response and substantial changes to the manuscript. I have no further comments.